# Technical note: Turbulence measurements from a Light Autonomous Underwater Vehicle

Eivind H. Kolås[1], Tore Mo-Bjørkelund[2], and Ilker Fer[1,3]

[1]Geophysical Institute, University of Bergen and Bjerknes Center for Climate Research, Bergen, Norway
[2]Norwegian University of Science and Technology, Trondheim, Norway
[3]Department of Arctic Geophysics, UNIS – The University Centre in Svalbard, Longyearbyen, Norway

**Correspondence:** Eivind H. Kolås (eivind.kolas@uib.no)

**Abstract.** A self-contained turbulence instrument from Rockland Scientific was installed on a Light Autonomous Underwater Vehicle (AUV) from OceanScan Marine Systems and Technology Lda. We report on the data quality and discuss limitations of dissipation estimated from two shear probes during a deployment in the Barents Sea in February 2021. The AUV mission lasted for 5 hours, operating at a typical horizontal speed of $1.1 \, \mathrm{m \, s^{-1}}$. The AUV was programmed to find and cross the maximum along-path thermal gradient at 10, 20 and 30 m depths along 4 km transects. Although the AUV vibrations contaminate the shear probe records, the noise is mitigated by removing vibration-induced components from shear spectra using accelerometer signal measured in multiple directions. Dissipation rate estimates in the observed transects varied in the range $1 \times 10^{-8}$ and $6 \times 10^{-6} \, \mathrm{W \, kg^{-1}}$, with the values from the two orthogonal probes typically in agreement to within a factor of 2. Dissipation estimates from the AUV show good agreement with nearby vertical microstructure profiles obtained from the ship during the transects, indicating that the turbulence measurements from the AUV are reliable for this relatively turbulent environment. However, the lowest reliable dissipation rates are limited to $5 \times 10^{-8} \, \mathrm{W \, kg^{-1}}$, making this setup unfit for use in quiescent environments.

## 1  Introduction

Turbulence measurements in the ocean are needed to quantify the turbulent fluxes of heat, salt and momentum, and are important for understanding the processes affecting evolution and transformation of water masses. The dissipation rate of turbulent kinetic energy provides the energy that homogenizes the gradients of temperature and salinity. Quantifying the magnitude and distribution of the dissipation rate helps identify the different forcing mechanisms and their relative contribution in mixing. In general, the necessary requirements for measuring ocean turbulence can be summed up in three elements: a sensor or probe that detects the physical parameter of interest; an electronic circuitry that amplifies and filters the signal produced by the probe; and a stable platform that is rigid or moves smoothly in the ocean (Lueck et al., 2002). Holding a probe stable while moving it smoothly through a dynamic ocean is not trivial. Ocean waves, currents and controlled platform adjustments will lead to platform motion and artificial signals not associated with natural turbulence.

The most common method for measuring ocean turbulence is to measure the small-scale velocity shear by using free-falling or loosely-tethered vertical microstructure profilers equipped with airfoil shear probes (Lueck, 2005; Gregg, 2021).

Such profilers are commonly deployed from vessels or drifting sea ice. However, the vertical profiling limits the horizontal and temporal resolution of the measurements. Robotic platforms offer the potential to increase the availability of ocean-mixing measurements (Frajka-Williams et al., 2022). Robotic platforms such as Autonomous Underwater Vehicles (AUVs) and ocean gliders enable turbulence measurements in a variety of patterns and detect structures that may be left undetected using vertical profiling alone (Yamazaki et al., 1990; Frajka-Williams et al., 2022).

Obtaining high-quality turbulence measurements from robotic platforms can be challenging. The vehicle motion, both forward motion and maneuvering, must be resolved and its effect on the measured signal must be filtered out. Pioneering work in the 90s used turbulence measurement packages and three orthogonal accelerometers mounted on AUVs, such as described by Levine and Lueck (1999) and Dhanak and Holappa (1999). Vehicle vibrations were found to completely obscure oceanic signals at distinct frequencies. Using coherency analysis between the shear probe record and the acceleration measured by the ac-

celerometer aligned with the shear probe, noise could be removed in the time and frequency domain (Levine and Lueck, 1999). Goodman et al. (2006) improved this technique and developed a multivariate correction approach to remove vibration-induced components from shear spectra using accelerometer signal measured in multiple directions. This latter way of minimizing the effects of body motion and probe vibrations on the turbulence measurements is commonly known as the "Goodman method", and paved the way for a range of robotic platforms with microstructure sensors.

Modern microstructure measurements using shear probes attached to robotic platforms include those from gliders (Fer et al., 2014; Palmer et al., 2015; Schultze et al., 2017; Scheifele et al., 2018), and from AUVs such as REMUS (Goodman et al., 2006) and the Autosub Long Range AUV (Thorpe et al., 2003; McPhail et al., 2019; Garabato et al., 2019; Spingys et al., 2021). Gliders are buoyancy driven and, in contrast to AUVs, do not use a thruster for forward motion (some new generation gliders can be equipped with a thruster for rapid maneuvering when needed). The smooth motion of the gliders with negligible

vehicle vibration and signal contamination makes them excellent platforms for shear probe measurements (Fer et al., 2014). While offering extended endurance of 1 to 3 months, gliders move relatively slow ($0.1$–$0.3\,\mathrm{m\,s^{-1}}$ through water) and typically profile in a saw-tooth pattern. AUVs move faster (order $1\,\mathrm{m\,s^{-1}}$ through water) and are more maneuverable, but have typically shorter endurance (order of hours to days). A major concern regarding the quality of turbulence measurements from an AUV, is the vibrations caused by the propulsion system.

In this study, we mounted a self-contained turbulence instrument package on a Light AUV, and collected measurements in the Barents Sea in a frontal region where Atlantic- and Arctic-origin waters meet (Figure 1). The Light AUV is lighter than REMUS and the Autosub Long Range AUV. With a typical configuration of sensors, it weighs about 35 kg in air, can be handled by one person, and enables easy deployment and recovery. In addition, the Light AUV is considerably more affordable compared to other AUVs, offers an open access software and ease of hardware configurations, making it a desirable and

versatile product.

In this technical note we describe the instrument setup and the data collected (Sect. 2), the processing methods (Sect. 3), and present the data quality and the capability of the Light AUV for dissipation rate measurements (Sect. 4). In our notation, data processing and format of the data, we follow the recommendations and conventions of the SCOR Working Group on

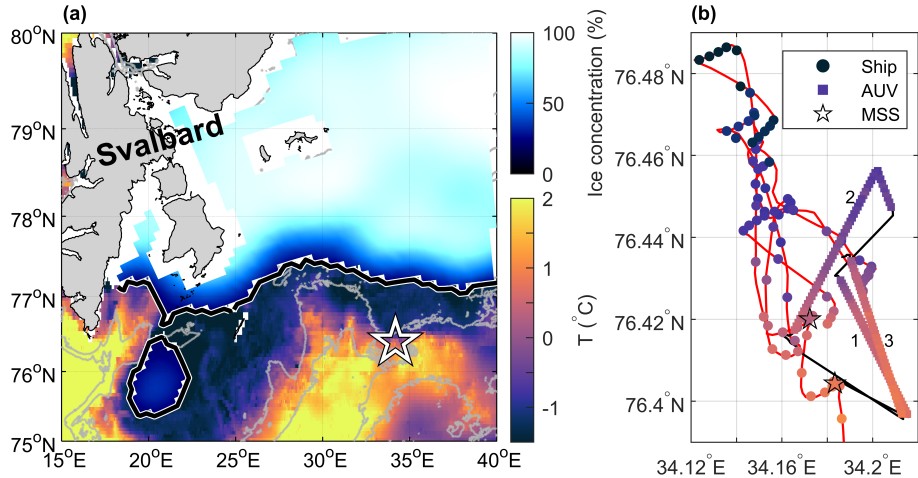

**Figure 1. a)** Overview map of the study region in the Barents Sea. Ice concentration and ice edge (thick black contour) on 26 February 2021 is from OSI SAF (OSI SAF, 2017). Sea surface temperature is from the product SEAICE_ARC_SEAICE_L4_NRT_OBSERVATIONS_011_008 at 0.05° resolution based upon observations from the Metop_A AVHRR instrument. Experiment location marked by a star near 34°E is expanded in (b). Grey isobaths are drawn at 200 and 300 m depth using IBCAO-v4 (Jakobsson et al., 2020). **b)** Ship's track (red) with near-surface temperature from the ship's thermosalinograph, and AUV's track (black) with AUV's temperature measurements along the three transects are color coded (temperature color scale is the same as in panel a). Stations where a vertical microstructure profile (MSS) was collected are also shown.

analysing ocean turbulence observations to quantify mixing (ATOMIX, http://wiki.uib.no/atomix). Data are available from Fer
et al. (2021).

## 2  Instruments, cruise and data

The data were collected during a Nansen Legacy cruise (9 February – 1 March 2021) on board the research icebreaker Kronprins Haakon, in the Barents Sea (Nilsen et al., 2021). Turbulence measurements using the Light AUV ("Harald", hereafter referred to as AUV) were made on the morning of 26 February 2021 near the Polar Front between Atlantic water and Polar water.
The AUV was deployed at 07:30 UTC at 76°N 24.94' 34°E 9.61', and recovered at 12:15 UTC at 76°N 26.11' 34°E 11.21', after completing three crossings of the front. Before and during the AUV mission, the wind speed was around $10\,\mathrm{m\,s^{-1}}$, air temperatures were close to -5°C measured at 15 m height, and the surface boundary layer extended to about 60 m depth. The turbulence package on the AUV continuously measured ocean microstructure. Additional data used include near-surface temperature and salinity measured by a Sea-Bird Electronics thermosalinograph with water intake at 4 m depth, and two
reference dissipation profiles measured by a vertical microstructure profiler (MSS-90L) from Sea and Sun Technology. The temperature and conductivity measured by the thermosalinograph are accurate to $\pm 0.001°$C and $\pm 0.001\,\mathrm{S\,m^{-1}}$. The noise level

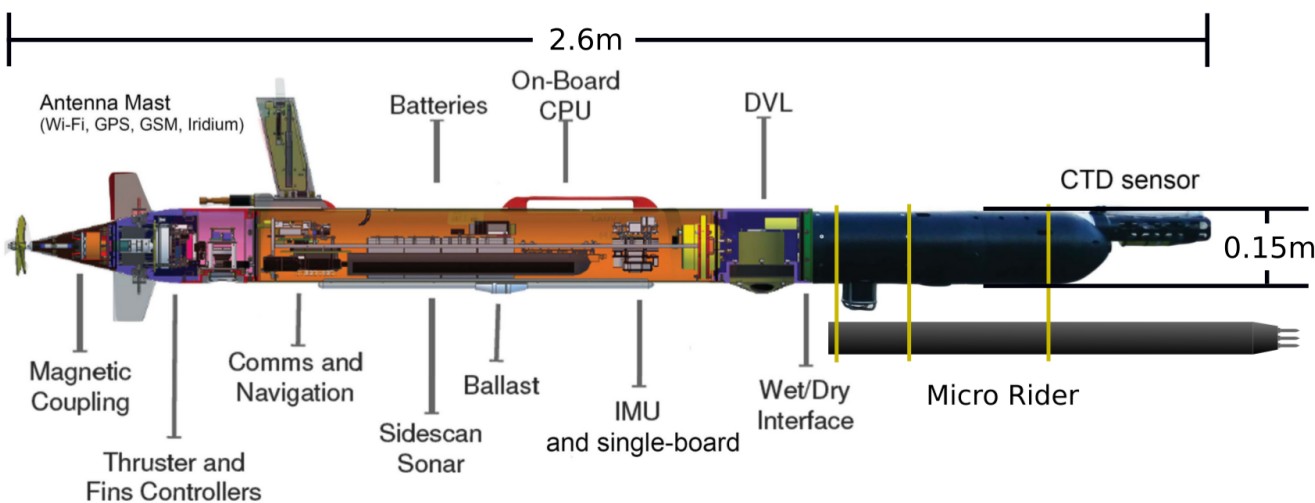

**Figure 2.** Sketch showing the AUV used on this mission, with instruments and hardware as indicated. This sketch is a modified version of a figure in Fossum et al. (2021).

of the dissipation measurements from the MSS-90L is $(1-3) \times 10^{-9}\,\mathrm{W\,kg^{-1}}$. The ship track, AUV track and MSS positions are shown in Figure 1b.

## 2.1 Light Autonomous Underwater Vehicle

The Light AUV was developed at the Underwater Systems and Technology Laboratory at the University of Porto (Sousa et al., 2012). It is commercially produced by OceanScan Marine Systems and Technology Lda. Our AUV (sketch shown in Figure 2) is an extended version compared to the standard Light AUV. It is 100 m pressure rated, and equipped with a pumped CTD (SBE-49 FastCAT), a Nortek Doppler Velocity Log (DVL1000), an attitude sensor (Lord Microstrain 3DM-GX4-25), an acoustic modem, a Fluorescence sensor and a dissolved oxygen optode. The accuracy of the measurements from the AUV

are $\pm0.002°\mathrm{C}$ for temperature, $\pm0.0003\,\mathrm{S\,m^{-1}}$ for conductivity, $0.3\%$ RMS (root mean square) of the measured value for horizontal flow speed past the instrument (measured by DVL1000), $\pm8.5°$ for yaw at the observation latitude and $\pm2.0°$ for pitch and roll. As the depth was about 250 m, the DVL1000 did not track bottom during this mission. The AUV trajectory was only constrained by inertial navigation, with an expected drift of about 15% of the distance traveled. The AUV is controlled by the on-board software DUNE Unified Navigation Environment, and is configurable both in hardware and software. The

expected mission duration is between a few hours to 48 hours, largely depending on the operating speed. While maximum speed can exceed $2\,\mathrm{m\,s^{-1}}$, a normal operating speed (without the turbulence package) is about $1.5\,\mathrm{m\,s^{-1}}$. The AUV communicates via satellite (iridium), WiFi, as well as acoustics. It can be remotely controlled within the WiFi range of about 200 m, which can be useful during deployment and recovery. While deployment is easily done from a ship using a crane (see Figure 3), recovery is best done from smaller workboats to avoid damaging the instrument. The turbulence package was mounted below the AUV

using custom-made brackets, and connected to the AUV using a bulkhead connector and a custom-made cable. Due to the extra

drag caused by the turbulence package, the operating speed during our mission was about $1.1\,\mathrm{m\,s^{-1}}$. Before deployment, we programmed the AUV to follow the frontal zone by tracking the maximum temperature gradient at different depths, which it successfully did.

## 2.2 Turbulence package

Turbulence measurements were made using a MicroRider-1000LP (MR) from Rockland Scientific, Canada. The MR was modified to the Tidal Energy (TE) configuration, earlier used in high flow tidal energy channels. The TE configuration includes increasing the sampling rate to 1024 Hz for fast channels (from the typical 512 Hz), and replacing the ASTP circuit board components with an anti-aliasing filter of 196 Hz (from the typical 98 Hz), and reducing the gain of the shear channel by a factor of 10, from about 1 second to 0.1 seconds. This modification allows reaching wavenumbers high enough to resolve the

shear spectrum (reaching 130 cpm at $1.5\,\mathrm{m\,s^{-1}}$ and with 196 Hz anti-aliasing filter). Reduction in the gain is to compensate for the larger signals produced by faster sensor speed through the water (the shear sensor signal increases in proportion to speed-squared).

The MR was attached beneath the AUV as seen in Figures 2 and 3. It was powered by a stand-alone 4S1P (14.8V) Lithium-Ion battery integrated into the vehicle, and controlled by a relay connected to the main power board inside the AUV. This was

done to provide a relatively clean power source. Earlier tests in a Norwegian fjord when the MR was fully integrated into the AUV power source, showed significant electronic noise in the microstructure measurements. Data were stored internally on a compact flash memory card. The vertical axis-to-axis separation between the AUV and the MR was approximately 30 cm. The flow field around AUVs with a similar shape and cross-section as the AUV we used, has been modeled (Mostafapour et al., 2018). Although this computational fluid dynamics modeling does not fully represent our AUV with the turbulence package

attached, it indicates the flow deformation around the AUV hull. All turbulence sensors protruded about 25 cm from the nose of the AUV, and are expected to sample flow with negligible deformation.

The MR was equipped with two airfoil velocity shear probes (SPM-38), one fast-response thermistor (FP07), a pressure transducer, a two-axis vibration sensor (a pair of piezo-accelerometers), and a high-accuracy dual-axis inclinometer. The MR samples the signal plus signal derivatives on the thermistor and pressure transducer, and the derivative for shear signals, allow-

ing high resolution measurements. The sampling rate is 1024 Hz for the vibration, shear and temperature sensors, and 128 Hz for pitch, roll and pressure. The accuracy of the measurements is 0.1% for the pressure, 2% for the piezo-accelerometers and 5% for the shear probes. Because of an error in the setup configuration file, the thermistor did not record measurements. Roll, pitch and yaw are clockwise rotations around the $x$, $y$ and $z$ axis of the AUV or the MR, following the right-hand rule. However, the instrument axis coordinate systems differ: for the MR $x$ points outward from the nose along the instrument's axis, $y$

is to the left (positive toward port) and $z$ is positive upward. For the AUV, the vehicle $xyz$-frame is aligned with [North, East, Down], $x$ is positive in the nominal vehicle direction of motion (forward), $y$ is to the right (starboard) and $z$ is positive in the down direction.

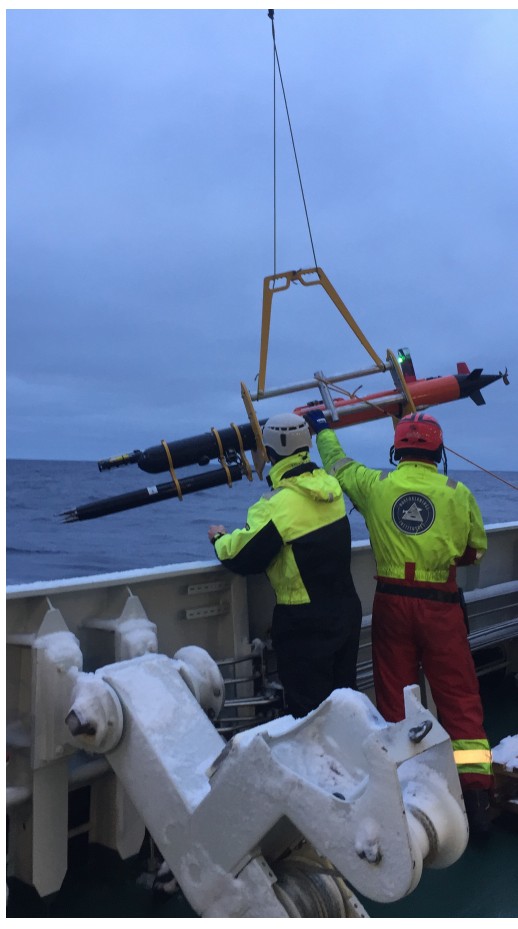

**Figure 3.** Deployment of MicroRider-1000LP mounted below the light AUV "Harald", in the Barents Sea, 07:30 on 26 February 2021. From left: Co-author Tore Mo-Bjørkelund and crew member Svein Are Simonsen. Photographer: Frank Nilsen, University Center in Svalbard

## 3 Processing

Before converting the raw data from the shear probes into physical units, the MR timestamp was corrected against the AUV timestamp. When the shear probe travels through the water horizontally along axis $x$, at speed $U$, then the voltage $E_p$ produced by the probe in response to a cross axis velocity $v$ is given by

$$E_p = 2\sqrt{2}\hat{s}Uv \ , \tag{1}$$

where the constant $\hat{s}$ is the sensitivity of the probe which must be determined by calibration (Lueck et al., 2002). The probe voltage is then converted to shear, $\partial v/\partial x$, in physical units as

$$\frac{\partial v}{\partial x} = \frac{1}{U}\frac{\partial v}{\partial t} = \frac{1}{2\sqrt{2}\hat{s}U^2}\frac{dE_p}{dt} \ , \tag{2}$$

by using the known sensitivity of the shear probe and the travel speed of the AUV (Lueck et al., 2002). The time derivative of $E_p$ is obtained from the differentiator in the electronics of the shear probe, with a known gain. A second probe oriented orthogonal to the first one similarly measures $\partial w/\partial x$. An initial high pass filtering at $0.6\,\mathrm{Hz}$ of the shear and vibration signals was performed in order to exclude signals at scales larger than the AUV (about 2 meters). Spectral loss due to high pass filtering was corrected for. In addition, both shear and vibration signals were despiked before calculating shear spectrum. Despiking was done by comparing the absolute shear and vibration time series to their $0.5\,\mathrm{Hz}$ low-passed records. When the ratio between the absolute and the low-passed time series exceeded 9 (8) for the shear (vibration), $\pm 0.04$ seconds centered at the spike were replaced by the value averaged over $\pm 0.5$ seconds before and after the spike.

Shear spectra are used to estimate the dissipation rate of turbulent kinetic energy, $\varepsilon$. The dissipation rate is proportional to the variance of shear contained at scales from $\mathcal{O}(1)\,\mathrm{m}$ to $\mathcal{O}(10^{-2})\,\mathrm{m}$. The time series from each shear probe was segmented into half overlapping 8-s long portions, corresponding to roughly $10\,\mathrm{m}$ portions along the transects. A fast Fourier transformation (FFT) length corresponding to 1 s was chosen, and each half overlapping 1-s segment was detrended, and smoothed using a Hanning window, before averaging them to get the shear frequency spectrum for each 8-s segment. Spectral loss due to the size of the shear probe was corrected for.

Shear spectra were converted from frequency, $f$, domain to wavenumber, $k$, domain using Taylor's frozen turbulence hypothesis and the AUV speed, $U$, as $k = f/U$. The Doppler Velocity Log (DVL) on the AUV measured $U$, and the average value for each 8-s segment was used in the conversion. Typical $U$ was $1.1\,\mathrm{m\,s^{-1}}$, thus the FFT length is equivalent to $1.1\,\mathrm{m}$ along-path length, and resolves the low wavenumber part of the spectrum while excluding scales greater than or equal to the vehicle length. For additional cleaning of the shear data, the shear spectrum signal coherent with the accelerometer spectrum signal was removed using the method described by Goodman et al. (2006).

Assuming isotropic turbulence, $\varepsilon$ was calculated for each segment by integrating the cleaned wavenumber spectrum, $\Psi(k)$ as

$$\epsilon = \frac{15}{2}\nu \overline{\left(\frac{\partial v}{\partial x}\right)^2} = \frac{15}{2}\nu \int_0^\infty \Psi(k)dk \approx \frac{15}{2}\nu \int_{k_1}^{k_u} \Psi(k)dk \tag{3}$$

where $\nu$ is the kinematic viscosity and overbar denotes averaging in time (e.g., Fer et al., 2014). The lower ($k_1$) integration limit is determined by the wavenumber corresponding to the FFT length, and the upper ($k_u < \infty$) integration limit is usually determined from a minimum in a low-order polynomial fit to the wavenumber spectrum in log-log space. Typically electronic noise takes over after the minimum in the spectrum. To account for the variance in the unresolved part of the spectrum (integration outside the $k_1$ and $k_u$ limits), the empirical model for turbulence spectrum determined by Nasmyth (1970) was used, hence the estimated $\varepsilon$ is close to the full integration. When using two shear probes, the dissipation rate in the segment is calculated as the average of the values from both sensors. In our data, the two sensors always agreed within a factor of 4.

From dissipation estimate time series, we extracted sections when the AUV performed horizontal transects at approximately constant depth with the propeller set to 1500 rotations per minute (RPM). During the horizontal transects the angle of attack (AOA), the difference between the pitch and the direction of travel, was much smaller ($< 3°$) than the critical value of $\pm 20°$ for

when the flow over the shear probe is no longer laminar (Osborn and Crawford, 1980). Final data screening excluded data with rate of change exceeding 10, 5 and 2 units per 1 s for roll, pitch and RPM, respectively. The thresholds in the final screening was determined from visual inspection of rate of change versus dissipation estimates.

## 4 Results

Five transects at depths 10, 20, 30, 40, and 50 m were planned across the temperature front; however, the mission ended abruptly after three transects due to a leak in the main hull of the AUV. The leakage was through the antenna and is a rare problem ($< 1\%$ of missions has ended due to leakages). Data recovered from the three transects are sufficient for the purpose of this technical note. Flight kinematics measured by the AUV is shown in Figure 4. Pitch was in general less than $2.5°$, and roll was less than $7.5°$. The relatively large average roll is probably a result of the positioning of the MR relative to the AUV, and the rolling moment induced by the propeller. The peak roll early in transect two is when the AUV made an abrupt turn (see figure 1b). Note that when the rate of change of roll, pitch and RPM was large, the dissipation rate data were excluded (Sect. 3). The propeller rate was set constant at 1500 RPM, yet the speed past the instrument varied between 1 and $1.2 \, \mathrm{m \, s^{-1}}$, seemingly related to the transition between the water masses (Figures 4b and c).

Figure 5a and b show mean shear spectra in frequency space using 8-s long records (length used for single dissipation estimates), for a moderate and a high value of $\varepsilon$, respectively. Corresponding vibration spectra from the accelerometers are also shown. The 95% confidence interval around a mean spectrum can be calculated as the factor $\exp(\pm 1.96 \times \frac{5}{4}(N_f - N_v)^{-\frac{7}{9}})$, where $N_f$ is the number of fft-segments and $N_v$ is the number of vibration signals (Lueck, 2022). Using $N_f = 15$ and $N_v = 2$, we obtain [0.72 1.40]. The 95% interval carried over to our epsilon estimates, including an additional 10% sensor sensitivity calibration uncertainty, becomes about [0.6 1.6].

An RPM of 1500 corresponds to 25 Hz, or using a mean speed of $1.1 \, \mathrm{m \, s^{-1}}$ to 23 cycles per meter (cpm). The contamination of the shear spectra by the propulsion system is visible in Figure 5; the propulsion system is not perfectly balanced around its rotational axis, and vibrations at 25, 50 and 75 Hz (and the harmonics of these frequencies) are induced by this off-center rotation. The main contaminating energy is at 75 Hz, related to the 3-bladed propeller. In addition, the accelerometers indicate that vibrations between 15 and 22 Hz also affect the shear signal; however, the source of these vibrations is not clear, and is discussed further in Section 5. The cleaned frequency spectra show that contamination from instrument vibration has been successfully removed and that the spectra resemble the empirical Nasmyth spectra (Nasmyth, 1970). Note, however, that the cleaned spectra also show a reduction in the spectral levels at low frequencies where the vibration signal is relatively low. The reduction in spectral levels and potential biases associated with the Goodman method are discussed in Section 5. The shear spectra in the wavenumber domain, for the same values of $\varepsilon$ as in (a) and (b), are shown in Figure 5c and d. In general, shear probe 1, $\partial w/\partial x$, resolves somewhat higher wavenumbers than shear probe 2, $\partial v/\partial x$. For moderate $\varepsilon$, shear probes 1 and 2 resolve wavenumbers up to 40 and 30 cpm respectively, while for high $\varepsilon$ they resolve wavenumbers up to 85 and 65 cpm, respectively. Beyond the resolved part of the spectrum, noise levels become too large, and the shear spectrum deviates significantly from the empirical Nasmyth spectrum.

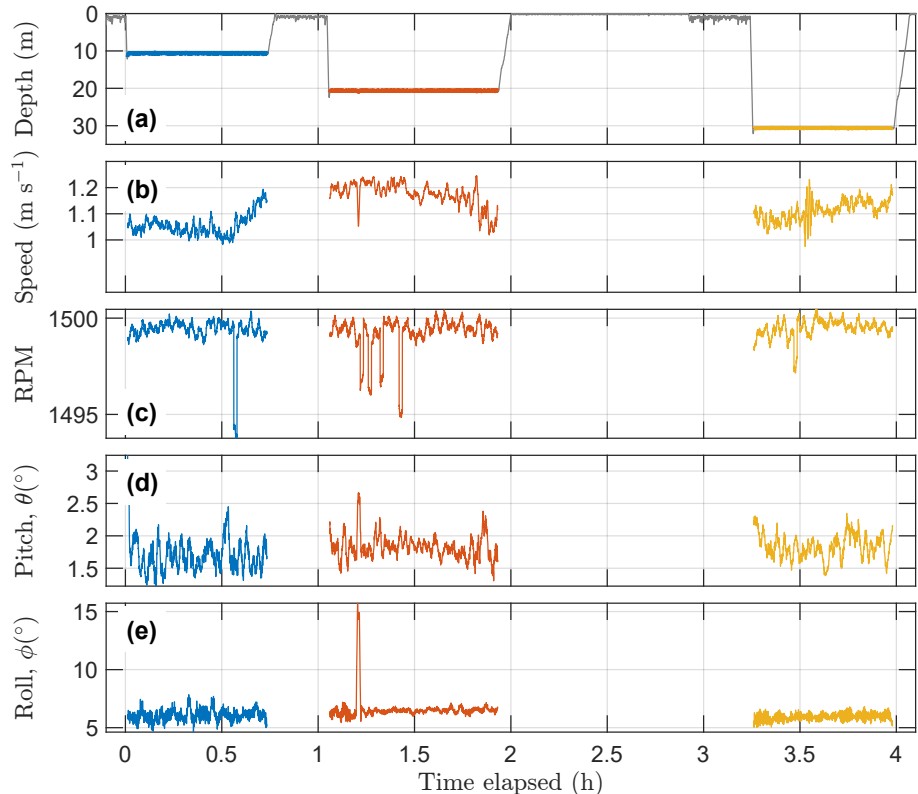

**Figure 4.** Flight kinematics from AUV. Time series of **a)** depth, **b)** speed past the instrument, **c)** rotation per minute (RPM), **d)** pitch and **e)** roll. Time elapsed is from 26 February 0806 UTC. Selected transects at approximately 10, 20 and 30 m are shown.

Further quality control of our data is done by bin-averaging $\varepsilon$ over different ranges. Figure 6a and b show bin-averaged clean spectra in wavenumber domain for $\partial w/\partial x$ and $\partial v/\partial x$, respectively. Values limiting the bins are listed in the caption. For $\varepsilon > 10^{-7}$ W kg$^{-1}$ the spectra closely resemble the bin-averaged Nasmyth spectra. However, the roll-off in the dissipation

200   subrange starts earlier than that indicated in the Nasmyth spectra, suggesting the most energetic dissipation rates are not fully resolved. For comparison, we also include the bin-averaged theoretical Panchev-Kesich spectrum (Panchev and Kesich, 1969) and observe that the roll-off of the Panchev-Kesich spectrum fits our cleaned spectrum better than the Nasmyth spectrum. Comparing $\partial w/\partial x$ to $\partial v/\partial x$, we observe that $\partial w/\partial x$ is generally capable of resolving wavenumbers 10-20 cpm higher than $\partial v/\partial x$. For $\varepsilon < 10^{-7}$ W kg$^{-1}$ the bin-averaged spectra start deviating from the empirical Nasmyth and theoretical Panchev-

205   Kesich spectra significantly for wavenumbers below 4 cpm, especially for $\partial v/\partial x$. While the difference in data quality delivered by the two probes is less than ideal, it is expected that the shear probes oriented orthogonally will sense the vehicle motion differently. Comparison with spectral shapes, vehicle motion, and noise sources are discussed further in Section 5.

The systematic difference in data quality seen in the low dissipation range in Figure 6, may manifest itself in the dissipation estimates. Figure 7 compares $\varepsilon_1$ and $\varepsilon_2$ calculated from the two different probes. The scatter plot (Figure 7a) shows that the two

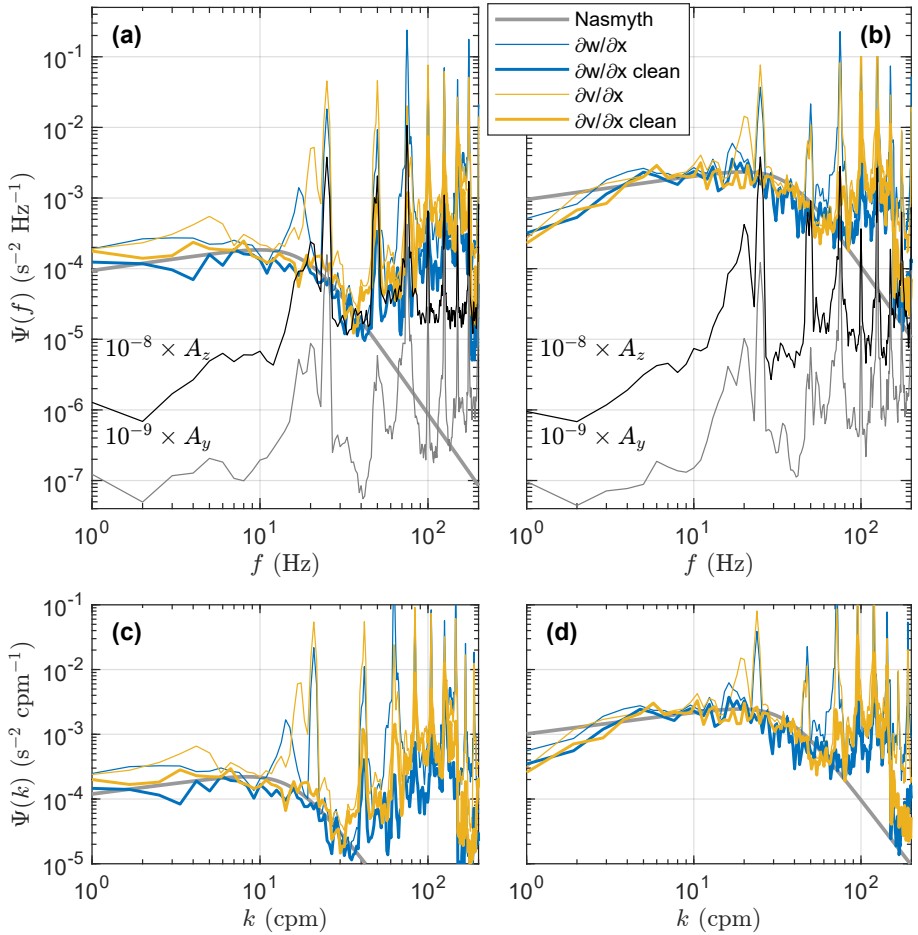

**Figure 5.** Example frequency spectra with **a)** moderate ($\varepsilon = 5.7 \times 10^{-8}\,\mathrm{W\,kg^{-1}}$), and **b)** high ($\varepsilon = 1.4 \times 10^{-6}\,\mathrm{W\,kg^{-1}}$) dissipation rates, using 8-s long records. Vibration spectra along the instrument's main and transverse axes are also shown with an offset as indicated. Cleaned spectra as indicated by the legend show frequency spectra after removing the shear probe signal coherent with accelerometer signal. Empirical Nasmyth spectra are shown for the values of $\varepsilon$. **c)** and **d)** show the same shear spectra as (a) and (b) respectively, but in the wavenumber domain. $\partial w / \partial x$ and $\partial v / \partial x$ are shear probes 1 and 2, respectively, on the MicroRider.

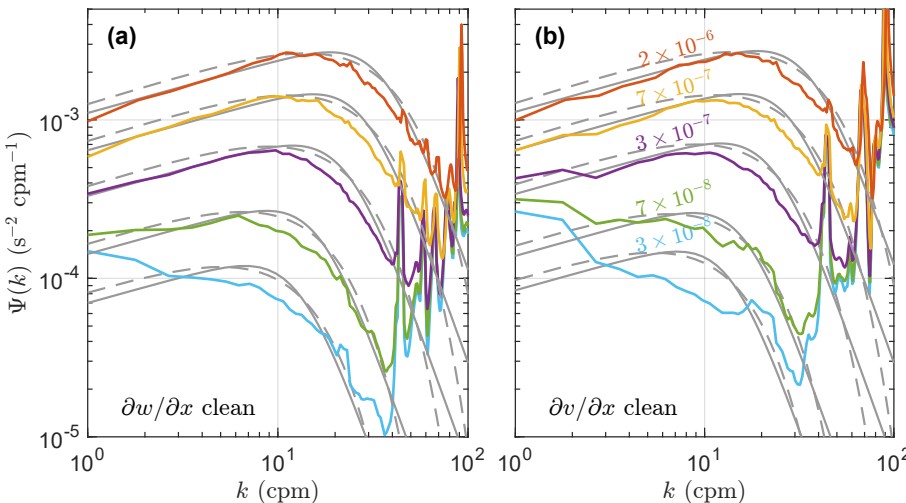

**Figure 6.** Wavenumber shear spectra of **a)** cleaned $\partial w/\partial x$ (shear probe 1), and **b)** cleaned $\partial v/\partial x$ (shear probe 2) averaged in increasing bins of dissipation rate estimates using data from all depths. Bin averaging limits are set to $1\times10^{-8}, 5\times10^{-8}, 1\times10^{-7}, 5\times10^{-7}, 1\times10^{-6}$ and $5\times10^{-6}$, averaging over 210, 145, 827, 466 and 382 ($\partial w/\partial x$) and 186, 146, 748, 529, 415 ($\partial v/\partial x$) spectra. Bin-averaged values of $\varepsilon$ with units in $W\,kg^{-1}$ are only shown in (b) as they were similar for both probes. Background curves are the bin-averaged (solid) Nasmyth and (dashed) Panchev-Kesich spectra, averaged over all individual estimates in the corresponding dissipation bins.

probes agree well within a factor of two. In fact, $97\%$ of the dissipation estimates from the two probes agree within a factor of two. Yet, while the disagreement between the two probes is more or less random for $\varepsilon > 5\times10^{-8}\,W\,kg^{-1}$, there is a systematic offset for $\varepsilon < 5\times10^{-8}\,W\,kg^{-1}$, where $\varepsilon_2$ shows higher dissipation rates than $\varepsilon_1$. The probability distribution function (PDF) for $\varepsilon_1$ and $\varepsilon_2$ (Figure 7b) show that the two probes in general agree very well for $\varepsilon > 10^{-7}\,W\,kg^{-1}$. For comparison of the three transects of the AUV, we show PDFs of $\varepsilon_1$ and $\varepsilon_2$ at 11, 21, and 31 m depth (Figure 7c, d and e, respectively). While the PDFs at 11 and 21 m depth resemble log-normal or skewed log-normal distributions where $\varepsilon_1$ and $\varepsilon_2$ typically agree (Fig. 7c, d), the PDF at 31 m depth differs. At this deeper transect, a larger portion of the $\varepsilon$ measurements is below $10^{-7}\,W\,kg^{-1}$. A second mode appears in low dissipation rates, particularly for $\varepsilon_2$, suggesting noise contributes significantly to the measurements at 31 m depth.

For additional quality control, we compare the final estimates of $\varepsilon$ to dissipation measurements from a vertical microstructure profiler (MSS-90 from Sea and Sun Technology), collected near the AUV transects during the AUV mission (see Figure 1b). The temperature sampled along the three transects at 11, 21, and 31 m depth, and the corresponding dissipation rates are shown in Figure 8a and b respectively. The dissipation rate of TKE varies throughout the different transects, but generally becomes smaller at greater depth, which is expected in the boundary layer. The arithmetic mean (including 95% confidence intervals) of the natural logarithm of the dissipation rates along the horizontal transects are compared to vertical microstructure profiles (Figure 8c). Although the spatial variability of $\varepsilon$ is known to be large (Yamazaki et al., 1990), the vertical profiles and the horizontal transects show comparable dissipation rates. Note however, that the comparison between the two MSS profiles

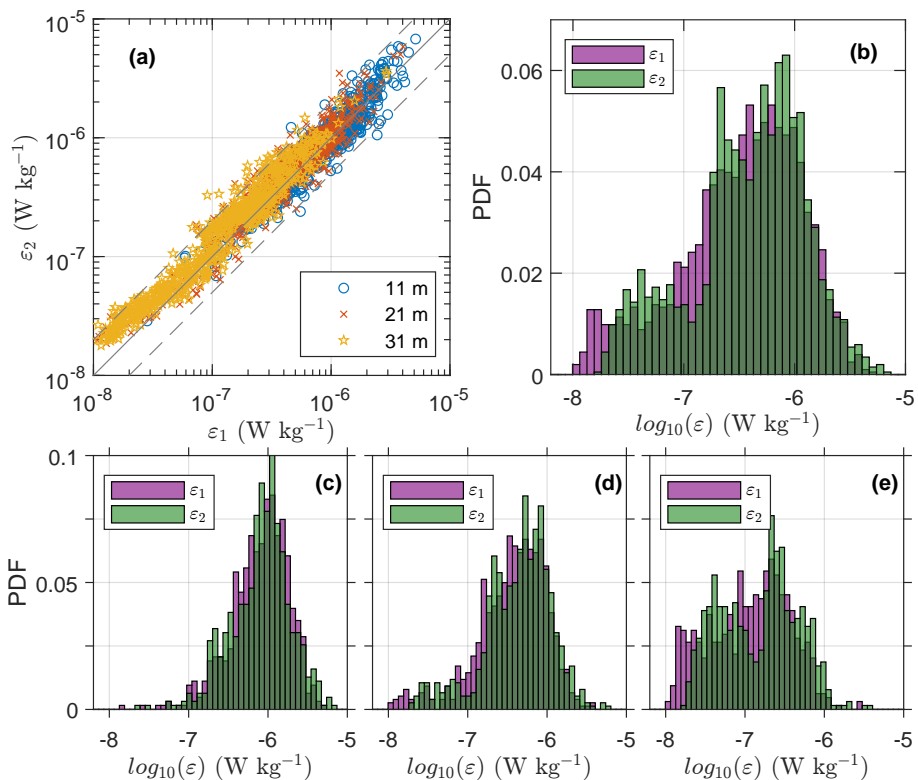

**Figure 7.** Comparison of dissipation estimates from two probes. $\epsilon_1$ is from $\partial w/\partial x$ measurements, and $\epsilon_2$ is from $\partial v/\partial x$. **a)** Scatter plot of dissipation estimates from each probe, color coded with respect to measurement depth. Gray dashed lines span the agreement within a factor of 2. **b)** Probability distribution function (PDF) for dissipation rates from each probe, using data from all depths. **c), d), e)** PDFs for dissipation rates from each probe, using data from transects at 11, 21 and 31 m depths, respectively.

and the average dissipation rates must be interpreted with caution. The two MSS profiles differ by one order of magnitude, enveloping the AUV-based measurements, and cannot be used to statistically test the validity of the AUV measurements.

## 5 Discussion

While microstructure measurements from gliders and larger AUVs have been extensively tested, microstructure measurements from smaller AUVs have not. Although the Light AUV is both more affordable and easier to handle than its larger siblings, it has potential drawbacks. Being smaller, the AUV is more susceptible to body motion and vibration, potentially contaminating the microstructure measurements. In addition, when the AUV platform is only a few times larger than the MR, AUV maneuvering skills may suffer from the added drag from the MR, depending on how the MR is integrated.

From Figure 5, we see that the shear spectra are significantly contaminated in the 10–30 Hz band (9–27 cpm), and in narrow bands centered at the integers of 25 Hz. The narrow-band peaks at 25, 50 and 75 Hz (and their higher harmonics) come

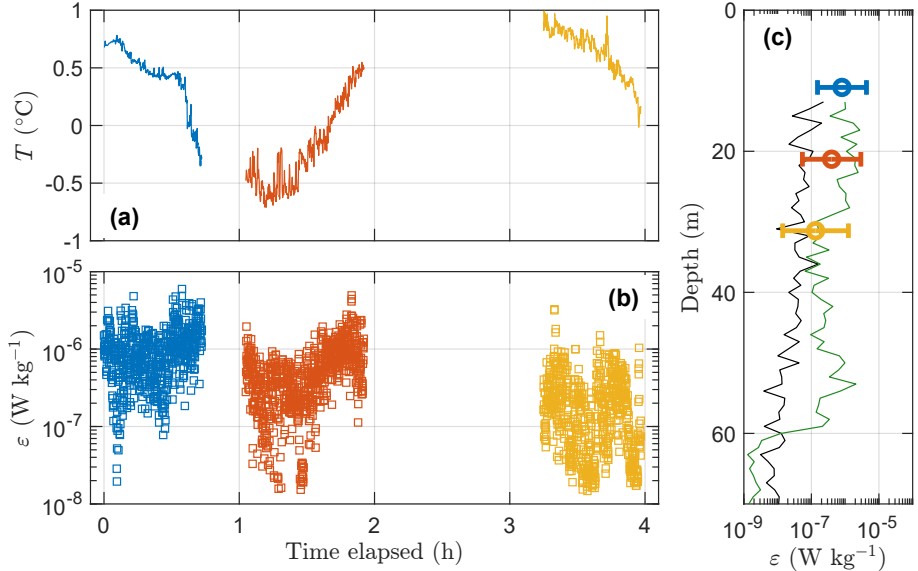

**Figure 8.** Overview of dissipation rates. Time series of **a)** temperature and **b)** final estimate of the dissipation rate. **c)** Vertical profiles (black and green) of $\varepsilon$ measured by the vertical microstructure profiler at two co-located stations marked by stars in Fig. 1b. Mean value and the lower and upper limits (95% confidence intervals) of the natural logarithm of the AUV-MR measurement are shown at their corresponding average depth in (c). Blue, red and yellow corresponds to 11, 21 and 31 m depth respectively.

from the three-bladed propeller operating at 1500 RPM. While vibrations from the propulsion system are less than ideal for turbulence measurements, the contamination occurs in a narrow band and is easily detected by both the accelerometers. The vibrations detected between 10 and 22 Hz (9–20 cpm) are more worrisome as this contamination covers a broader band of the

turbulence spectrum, in the wavenumbers where the spectrum typically rolls off. The spectral peaks in the shear spectra are at different frequencies for $\partial w/\partial x$ and $\partial v/\partial x$, suggesting that the vehicle motion (pitch, roll, yaw) is the main source of this contamination. While $\partial v/\partial x$ will be affected by the roll and yaw fluctuations, $\partial w/\partial x$ will be affected by changes in pitch but should be fairly insensitive to changes in roll and yaw.

The method for noise removal relies on the squared-coherency between the shear probe signal and the accelerometer signals

(Goodman et al., 2006). Removal of the shear probe signal coherent with the accelerometer signal produces clean spectra (Figure 5). The clean spectra show that the spikes in the 10-30 Hz band have been successfully removed. Note, however, that squared-coherency will always be nonzero even when the shear probe and accelerometer time series are completely incoherent. This results in a bias by removing some of the incoherent signal. We did not correct this bias in our study. When we apply a simple correction recommended in the ATOMIX guidelines, the clean spectra increase by a factor of about 1.2. This potential

bias does not affect our conclusions.

The bin-averaged shear spectra suggest that the most energetic wavenumbers are not fully resolved by our instrumentation; i.e., the transition between the inertial subrange and the dissipation subrange rolls off at lower wavenumbers compared to the

similarly bin-averaged Nasmyth and Panchev-Kesich spectra (Figure 6). While the shape of the spectral roll-off is relatively similar to that in the Panchev-Kesich spectrum, the offset between the observed and the theoretical spectra is significant. The

bias unaccounted for in application of the Goodman method cannot explain this offset fully. Some of the discrepancies in the roll-off are likely caused by averaging the spectra over variable dissipation rates, where the spectral peak shifts to higher wavenumbers with increasing $\varepsilon$, which will smooth the spectral roll-off. Yet, to mimic the effect of smoothing of the spectral roll-off, we bin-average individual Nasmyth and Panchev-Kesich spectra similarly. Another possible reason for the difference between the observed and the Nasmyth spectra is that an unknown fraction of the removed shear probe signal coherent with

the accelerometers can be natural turbulence, indistinguishable from the vibrations caused by the AUV (Palmer et al., 2015). This likely leads to a reduction of variance in the contaminated band between 10–30 Hz (9–27 cpm).

The average shear spectra for the smaller dissipation rates (Figure 6) deviate from the Nasmyth shape for small wavenumbers. Combined with the issues resolving the spectral roll-off, this suggests the instrument is not able to resolve dissipation rates smaller than about $5 \times 10^{-8}$ W kg$^{-1}$. This is particularly problematic for $\partial v / \partial x$ ($\varepsilon_2$). For $\varepsilon < 5 \times 10^{-8}$ W kg$^{-1}$, there is a

systematic offset between the two shear probes in the low wavenumber part of the spectrum (< 3 cpm). In weakly-turbulent regimes, the assumption of local isotropy may be violated, and the dissipation estimates from the orthogonal probes deviate when the buoyancy Reynolds number ($\frac{\varepsilon}{\nu N^2}$) is about 200 or less (Yamazaki and Osborn, 1990). Here, the AUV mission is conducted within the weakly-stratified upper surface layer with large buoyancy Reynolds numbers ($> 10^4$, not shown), and we do not expect differences caused by vertical stratification or anisotropy at probe separation scales. As with the noise

contamination in the 10-22 Hz band, the difference between the two probes is likely because the two probes sense the changes in pitch, roll and yaw differently. Furthermore, the effect of pitch, yaw and roll on the shear sensors is also dependent on how the MR is mounted on the AUV.

When mounting the MR on the AUV, our main concern was to ensure that the shear sensors protruded outside the region of flow deformation, without modifying the AUV itself. To avoid interfering with the acoustic modem and fluorescence sensor

on the upper part of the AUV, we mounted the MR below the AUV using brackets. This solution led to unwanted pitching at higher velocities due to the change in the center of drag. An alternative solution would be to re-design the wet-section (nose) of the AUV to fit the MR. This would likely lead to better AUV maneuverability, reducing changes in pitch, roll and yaw, hence reducing the vehicle motion sensed by the MR.

The MR was modified to the Tidal Energy (TE) configuration (Sect. 2.2), to allow for sufficiently resolved measurements

at high operation speeds of the AUV. The AUV used in this paper has the capability of moving at speeds exceeding 2 m s$^{-1}$. The practical application in this study limited the maximum operating speeds to about 1.5 m s$^{-1}$ because of the drag added by the MR. To further limit vibrations, we kept the operation speed at $1 - 1.2$ m s$^{-1}$. For such operating speeds, the standard MR configuration could work satisfactorily. However, with a better integrated MR, for instance inside the wet-nose section of the AUV, higher speeds would be achievable with reduced drag, necessitating the use of the TE configuration.

# 6 Summary and Conclusions

A modified MicroRider-1000LP was mounted below a Light AUV and tested in the Barents Sea during a cruise in February 2021. The AUV conducted three transects across a surface temperature front at 11, 21 and 31 m depth, while continuously sampling microstructure shear. The dissipation rate of turbulent kinetic energy is estimated from the shear measurements. Although the vibrations of the AUV contaminate the shear probe records, the shear spectra for dissipation levels above $5 \times 10^{-8}$ W kg$^{-1}$ are sufficiently cleaned using the Goodman method (Goodman et al., 2006). Dissipation rates measured from the AUV agree well with the measurements using a loosely-tethered vertical microstructure profiler from the ship. However, the overall noise level from the AUV is quite large; this setup cannot detect dissipation rates below $5 \times 10^{-8}$ W kg$^{-1}$ reliably, and is unfit for use in quiescent boundary layers. An improved installation of the turbulence probes on the nose of the AUV could reduce some of the limitations reported here and allow acceptable quality dissipation measurements from the AUV in relatively quiet environments.

*Data availability.* The AUV and MicroRider data set is available from Fer et al. (2021) through the Norwegian Marine Data Centre, https://doi.org/10.21335/NMDC-1821443450 with a Creative Commons Attribution 4.0 International License. SST data are obtained from the EU Copernicus Marine Service Information, product SEAICE_ARC_SEAICE_L4_NRT_OBSERVATIONS_011_008.

*Author contributions.* IF, TM-B and EHK collected the data, conceived and planned the analysis. IF and EHK performed the analysis. EHK wrote the paper, with advice and critical feedback from IF and TM-B. All authors discussed the results and finalized the paper.

*Competing interests.* Authors have no competing interests. IF is a member of the editorial board for the Ocean Science

*Acknowledgements.* The research was funded by the Research Council of Norway through the Nansen Legacy project (276730). We thank the officers and crew of the Kronprins Haakon for their skillful operations, the cruise leader Frank Nilsen for supporting the experiment. Martin Ludvigsen facilitated the AUV and provided valuable advise in preparations and planning. We thank Rolf Lueck and Evan Cervelli at Rockland Scientific for their advice and assistance in modifying the MicroRider for the AUV application. The Nansen Legacy uses NIRD as data depository (account numbers NS9610K and NS9530K). Figure 1a is produced using E.U. Copernicus Marine Service Information.

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
