# Peer review of "Technical note: Turbulence measurements from a Light Autonomous Underwater Vehicle"

_Ocean Science, 2021_

## Author Response (AR1)

**Response to referee comment on "Technical note: Turbulence measurements from a Light Autonomous Underwater Vehicle" by Eivind Hugaas Kolås et al., Ocean Sci. Discuss., https://doi.org/10.5194/os-2021-107-RC1, 2021**

We thank both reviewers for their constructive comments and useful suggestions, which helped to improve the manuscript. Below we provide a point-by-point response to all comments raised by the reviewers. Reviewers' comments are reproduced in *italic type in red* followed by our response in regular type, black color.

**Response to Reviewer 1**

*This paper reports on experiences with integrating microstructure shear sensors on a lightweight autonomous underwater vehicle (AUV). A particular challenge to overcome is that small AUV fly less stably, being actively propelled and maneuvering, contaminating microstructure spectra. The authors report good data quality down to about 5e-8 W/kg dissipation, which makes it fit only for measurements in turbulent boundary layers, which I do not see as too big a constraint given the lightweight nature of the AUV in the first place. This technical note is very thorough, and I only have a few comments and further questions that the authors may wish to consider during revision.*

Thank you for the positive comments.

*75: DVL was downward looking? DVL1000, I assume means a 1 MHz instrument? With 1MHz the DVL probably did not have bottom lock (Though I have not checked the bottom depth at your location) - how is the trajectory (see Fig 1b) so well-constrained? What is the navigation accuracy? Were there other navigation aids apart from inertial?*

Indeed, the DVL1000 is a downward-pointing 1MHz instrument. It did not have bottom lock during this mission as the depth was about 250 m. The trajectory is only constrained by inertial navigation, and the expected drift is about 15% of distance travelled.
We now added two sentences explaining this:

> pitch and roll. As the depth was about 250 m, the DVL1000 did not track bottom during this mission. The AUV trajectory was
> only constrained by inertial navigation, with an expected drift of about 15% of the distance traveled. The AUV is controlled
> 85 by the on-board software DUNE Unified Navigation Environment, and is configurable both in hardware and software. The

*147: Why is this used as opposed to the methodologies used by Moum et al. 1995 or Fer 2006 (iterative integration)?*

Thanks for pointing out this inaccurate processing description. The upper integration limit is set by the smallest of several criteria that are explained thoroughly at the ATOMIX wiki page https://wiki.uib.no/atomix/index.php/Iterative_spectral_integration_algorithm (see a snapshot below). The 5th criterion described here is the iterative integration. Yet for most cases, the limiting factor is constrained by the electronic noise because the integration limits from other criteria result in typically larger wavenumbers than where the noise starts to dominate the shear spectrum. This is determined by fitting a low-order polynomial (the 1st

criterion). We now rewrite this sentence as; "…and the upper $(k_u < \infty)$ integration limit is usually determined from a minimum in a low-order polynomial fit to the wavenumber spectrum in log-log space.".

Description from the ATOMIX wiki:

The upper limit of integration, $k_u$, is set by the smallest of a number of criteria that are listed next.

(1) The wavenumber range that is dominated by electronic noise can usually be determined from a minimum in the spectrum. Real shear is at wavenumbers smaller than this minimum while electronic noise is usually at higher wavenumbers. The spectral minimum may be found by fitting a low-order polynomial to the spectrum in log-log space. Third order is often sufficient. The wavenumber of the spectral minimum, $k_{\min}$, sets one of the limits on $k_u$.

(2) Another upper limit is $k_{150} = 150\,\mathbf{cpm}$ that is imposed by the spatial resolution of a commonly used shear probe. You may use a different value if your shear probe has a spatial resolution different from that reported by Macoun and Lueck, 2004[2] At the wavenumber of $150\,\mathbf{cpm}$ the spectrum derived from the commonly used shear probe is boosted by a factor of 10. At higher wavenumbers the spectral correction is more than a actor of 10 and such large corrections are not recommended.

(3) The cut-off frequency, $f_A$, of the anti-aliasing filter used by the shear-probe sampler sets another upper limit of spectral integration, namely $k_A \le f_A/U$. Because most filters have a transition range from passing to attenuating a signals, it is wise to set this limit to value slightly smaller than the cut-off frequency. For example, $k_A \le 0.9\,f_A/U$.

(4) The user may impose an upper limit to exclude wavenumbers that have contaminations that are not correctable, $f_{\lim}$. For most instruments this limit is usually set to $\infty$, but it may be prudent to set this limit to a finite value in some cases.

(5) The final wavenumber limit, $k_{95}$, is the wavenumber at which the variance of shear is resolved to 95%. There is not incentive to integrate the spectrum beyond this limit because the correction that must be applied amounts to only 5%. This wavenumber is $k_{95} = 0.12\,(\varepsilon/\nu^4)^{1/4}$ and the factor of $0.12$, is nearly identical for all of the common approximations to the shear spectrum, such as the approximations to the Nasmyth [3] [4] [5], the Panchev-Kesich [6], and the Lueck [5] non-dimensional universal spectra.

Thus, the upper limit of spectral integration is

$k_u = \min(k_{\min},\ k_{150},\ 0.9k_A,\ k_{\lim},\ k_{95})$.

The last of these upper limits, $k_{95}$, presets us with a conundrum because it requires the rate of dissipation which is what we are trying to estimate by way of the integration of the shear spectrum. Clearly, we need to bootstrap this process by starting with a reasonable (but certainly a rough) estimate of the rate of dissipation.

*184 and 230, regarding data quality differences between dw/dx and dv/dx: You could consider adding a remark on how stratification and/or violations of isotropy may play a role or not, given that it seems to be worse at lower dissipation (lower buoyancy Reynolds number - but how low?).*

Thank you for suggesting this. From the two vertical microstructure profiles presented, we obtain large values of the buoyancy Reynolds number (>100) for the entire water column, and very large in the upper 60 m (>10$^4$). This is consistent with the estimates from the LAUV dissipation measurements using the buoyancy frequency calculated at the measurement level. It is unlikely that the difference between the two probes is caused by vertical stratification or anisotropy at probe separation scales. We now add this information to the discussion.

> systematic offset between the two shear probes in the low wavenumber part of the spectrum (< 3 cpm). In weakly-turbulent regimes, the assumption of local isotropy may be violated, and the dissipation estimates from the orthogonal probes deviate when the buoyancy Reynolds number ($\frac{\varepsilon}{\nu N^2}$) is about 200 or less (Yamazaki and Osborn, 1990). Here, the AUV mission is conducted within the weakly-stratified upper surface layer with large buoyancy Reynolds numbers ($> 10^4$, not shown),
>
> 270 and we do not expect differences caused by vertical stratification or anisotropy at probe separation scales. As with the noise

**Response to Reviewer 2**

*"Technical note: Turbulence measurements from a Light Autonomous Underwater Vehicle"*
*by Eivind H. Kolås, Tore Mo-Bjørkelund, and Ilker Fer*

*The authors report on a successful test deployment of a small AUV combined with a turbulence microrider. They show that issues arising from thruster vibrations can be overcome in high mixing environments, by post processing. The resulting turbulent dissipation rates are plausibly reliable (while uncertainty limits have not been estimated). The paper is written in a clear manner and will fit the journal, after a number of necessary additions and revisions.*

Thank you for the positive and constructive feedback. Following both reviewers' comments, we provided substantial revisions and clarifications that improved the previous version of the manuscript.

*The program of the paper is well distilled in the title and the first two sentences of the abstract (technical note; combination AUV - MR; data quality), so that it seems natural to structure the major remarks accordingly:*

*(1) aspects of the combination lightAUV - MR: pro and contra, future potential.*

*(2) aspects of a technical note: complete and precise descriptions and practical hints.*

*(3) aspects of data quality to be expected.*

Your structured and thorough comments have been very helpful and are much appreciated. Below we reply to each of your remarks.

*(1)*

*The paper does not talk much about why the combination of light AUV and MR was used, or why other researchers should take this step. There is one passage in the course of the paper: the light AUV is lighter, cheaper, easier to handle than heavy AUVs (L.53f). I'd recommend to inform potential users in a condensed short discussion (might be in the Introduction or Conclusion): what past gap in turbulence measurements can be closed by this system; what is its future potential; why and when should a researcher owning a glider-MR or heavyAUV-MR switch to lightAUV-MR?*

We agree with the reviewer. We now include a short description in the introduction about the potential advantages of this system. In addition, we elaborate on the potential usages of this system in the discussion.

> In this study, we mounted a self-contained turbulence instrument package on a Light AUV, and collected measurements in the Barents Sea in a frontal region where Atlantic- and Arctic-origin waters meet (Figure 1). The Light AUV is lighter than REMUS and the Autosub Long Range AUV. With a typical configuration of sensors, it weighs about 35 kg in air, can be handled by one person, and enables easy deployment and recovery. In addition, the Light AUV is considerably more affordable
> 55 compared to other AUVs, offers an open access software and ease of hardware configurations, making it a desirable and versatile product.

*What are the constraints of this system? Is it really 100m depth range, order 10 hours of mission duration, 1m/s speed, cold water environment, high turbulence environment, short-distance communication only via acoustic modem, epsilon uncertainty of factor 2 to 5; as might be inferred from different parts of the paper? Are these final constraints or is there potential for pushing the limits?*

The LAUV system is constrained to 100 m depth (although the MR is 1000-m capable). The mission duration largely depends on the operating speed and is usually around 12 h. Another version of the LAUV with a larger battery that can last up 72 h is also available. Speeds can reach 4 knots (about 2 m/s), yet the normal operating speed is about 1.5 m/s (without the MR). The water temperature is not a constraint. This particular MR/LAUV setup is only suitable for high turbulence environment, but improvements on the MR integration can potentially reduce the background noise contamination. In our data, 97% of dissipation estimates from the two sensors agree within a factor of 2. Communication can be done over the horizon via satellite (Iridium), which was the case in our study. Short-distance communication can be done through Wi-Fi and acoustics.

We now include more details on the LAUV constraints in Section 2.1 (see the snapshot from the revised manuscript in response to the next comment). In addition, in the discussion section, we elaborate more on the limitations and potentials.

*(2)*

*There are some open technical questions that users probably would be interested in.*
For the following comments to section 2.1, we now include more details as suggested. A snapshot of section 2.1 is inserted below. We also answer each comment separately below.

75 **2.1    Light Autonomous Underwater Vehicle**

The Light AUV was developed at the Underwater Systems and Technology Laboratory at the University of Porto (Sousa et al., 2012). It is commercially produced by OceanScan Marine Systems and Technology Lda. Our AUV (sketch shown in Figure 2) is an extended version compared to the standard Light AUV. It is 100 m pressure rated, and equipped with a pumped CTD (SBE-49 FastCAT), a Nortek Doppler Velocity Log (DVL1000), an attitude sensor (Lord Microstrain 3DM-GX4-25),
80    an acoustic modem, a Fluorescence sensor and a dissolved oxygen optode. The accuracy of the measurements from the AUV are $\pm 0.002°$C for temperature, $\pm 0.0003\,\mathrm{S\,m^{-1}}$ for conductivity, 0.3% RMS (root mean square) of the measured value for horizontal flow speed past the instrument (measured by DVL1000), $\pm 8.5°$ for yaw at the observation latitude and $\pm 2.0°$ for pitch and roll. As the depth was about 250 m, the DVL1000 did not track bottom during this mission. The AUV trajectory was only constrained by inertial navigation, with an expected drift of about 15% of the distance traveled. The AUV is controlled
85    by the on-board software DUNE Unified Navigation Environment, and is configurable both in hardware and software. The expected mission duration is between a few hours to 48 hours, largely depending on the operating speed. While maximum speed can exceed $2\,\mathrm{m\,s^{-1}}$, a normal operating speed (without the turbulence package) is about $1.5\,\mathrm{m\,s^{-1}}$. The AUV communicates via satellite (iridium), WiFi, as well as acoustics. It can be remotely controlled within the WiFi range of about 200 m, which can be useful during deployment and recovery. While deployment is easily done from a ship using a crane (see Figure 3), recovery

90     is best done from smaller workboats to avoid damaging the instrument. The turbulence package was mounted below the AUV using custom-made brackets, and connected to the AUV using a bulkhead connector and a custom-made cable. Due to the extra drag caused by the turbulence package, the operating speed during our mission was about $1.1\,\mathrm{m\,s}^{-1}$. Before deployment, we programmed the AUV to follow the frontal zone by tracking the maximum temperature gradient at different depths, which it successfully did.

*For section 2.1:*

*- What mission duration can I expect? (battery endurance? reliability? see leak after 5 hours)*
Mission duration can be between a few hours to 48 hours with the standard battery, depending on the operating speed (1-2.6 m/s). The version with extended battery capacity can last up to 72 hours.
In this LAUV's lifetime, there have been three leakages from a total of about 1000 deployments. The cause of the leakages was a batch of bad antennas.

*- Is remote control during the mission needed/possible at all? How? At what range?*
Remote control is possible when the vehicle is at the surface and within Wi-Fi-range (200 m from the mothership). It can be remotely controlled using a smartphone (need to download an application). It is not necessary to remotely control it, but it can be useful during deployment and recovery.

*- Size/weight of the AUV (Fig.2 can only give a hint)?*
Length: 2.6 m, diameter: 15 cm, weight: 35 kg.

*- How is the handling and constraints for deployment and, particularly, recovery?*
Deployment can be done from a crane on board the mothership, however, one needs to recover the vehicle from a workboat or similar. This adds constraints to the weather conditions.

*- Is there light AUV versions with higher pressure rating?*
Not currently, but a 200 m version is in the making.

*- Particulars about the MR mounting, possibly a drawing.*

The following figure illustrating the setup has now been included in the paper.

[Figure]

*- Is there anything known about self-oscillation of the system (maybe from the MR vibration sensor while thruster off)?*
No, there is unfortunately no information about the self-oscillation of the system available.

*For section 2.2:*

*- How severe would have been the consequences, if the MR had not been in TE configuration? Referring to Fig.4 it seems that resolving wavenumbers up to 82 cpm instead of 164 cpm could still work to some degree.*
We agree with the reviewer that in this particular dataset, the standard MR configuration could work satisfactorily to some degree. The LAUV used in this paper has the capability of moving at speeds exceeding 4 knots (>2 m/s). Due to the drag of the MR mounted below the AUV, the actual maximum operating speeds were limited to about 1.5 m/s. However, as higher speeds generally caused more artificial vibrations and vehicle oscillation, we decided to not go faster than about 1.2 m/s. We agree that for operating speeds no more than about 1 m/s, the standard MR configuration may have been sufficient. Yet, with a better integrated MR (for instance a MR mounted inside the wet-nose section of the AUV) higher speeds would be achievable with reduced drag. We now elaborate on operating speeds in section 2.1, and states in section 2.2 (caption below) that these modifications allow reaching wavenumbers up to 130 cpm at 1.5 m/s. We also added a paragraph in the discussion about the operating speed and TE configuration (see snapshots below).

2.2:

100    factor of 10, from about 1 second to 0.1 seconds. This modification allows reaching wavenumbers high enough to resolve the shear spectrum (reaching 130 cpm at $1.5 \, \mathrm{m\,s^{-1}}$ and with 196 Hz anti-aliasing filter). Reduction in the gain is to compensate

Discussion:

280    The MR was modified to the Tidal Energy (TE) configuration (Sect. 2.2), to allow for sufficiently resolved measurements at high operation speeds of the AUV. The AUV used in this paper has the capability of moving at speeds exceeding $2 \, \mathrm{m\,s^{-1}}$. The practical application in this study limited the maximum operating speeds to about $1.5 \, \mathrm{m\,s^{-1}}$ because of the drag added by the MR. To further limit vibrations, we kept the operation speed at $1 - 1.2 \, \mathrm{m\,s^{-1}}$. For such operating speeds, the standard MR configuration could work satisfactorily. However, with a better integrated MR, for instance inside the wet-nose section of the

285    AUV, higher speeds would be achievable with reduced drag, necessitating the use of the TE configuration.

*- (L.106) Is there more known about the flow deformation around the system?*
The flow field around AUVs with similar shape and cross-section as the LAUV we used, has been modeled (Mostafapour et al. 2017; DOI: 10.29252/jafm.11.02.28302). Although this cannot fully represent our case, their figure 4 shows the flow deformation around the AUV hull. According to the velocity contours, the shear probes protrude sufficiently away from the body of the AUV and are expected to sample flow with negligible deformation. The additional flow deformation due to protruding instruments (such as the CTD and MR) is unknown. The citation and clarification are inserted in section 2.2.

compact flash memory card. The vertical axis-to-axis separation between the AUV and the MR was approximately 30 cm. The flow field around AUVs with a similar shape and cross-section as the AUV we used, has been modeled (Mostafapour et al.,

110    2018). Although this computational fluid dynamics modeling does not fully represent our AUV with the turbulence package attached, it indicates the flow deformation around the AUV hull. All turbulence sensors protruded about 25 cm from the nose of the AUV, and are expected to sample flow with negligible deformation.

*- (L.128) Has the spectral loss due to the high pass filter been corrected during post processing? And akin to this question, has the spectral loss due to the finite size of the airfoils been corrected?*

Yes, both have been corrected for. The high-pass filter correction is obtained by multiplying the spectrum by $1 + (f_c/f)^2$ where $f_c$ is the high-pass cut-off frequency and $f$ is the frequency. This correction only affects the spectrum near the cutoff frequency of the high-pass filter, $f_c$. The correction due to the size of the shear probe is done by multiplying the spectra by the factor $1 + \left(\frac{k}{k_0}\right)^2$ where $k_0$ = 50 cpm and $k$ is the wavenumber expressed in units of cpm.

We now clarify these corrections in section 3 (lines 135-136 and lines 144-145 shown below).

135    was performed in order to exclude signals at scales larger than the AUV (about 2 meters). Spectral loss due to high pass filtering

    was corrected for. In addition, both shear and vibration signals were despiked before calculating shear spectrum. Despiking

    Hanning window, before averaging them to get the shear frequency spectrum for each 8-s segment. Spectral loss due to the size

145    of the shear probe was corrected for.

*- (L.130) How is despiking done?*

The initial despiking is done by comparing the absolute shear and vibration time series to a 0.5 Hz low passed time series. When the ratio between the absolute and the low passed time series of shear or vibration exceeds a set threshold (a value of 8 typically works fine), the spike +-0.04 s is replaced by the average value over 0.5 s before and after the spike.

We now include these details in the processing description (lines 136-139 shown below)

    was corrected for. In addition, both shear and vibration signals were despiked before calculating shear spectrum. Despiking

    was done by comparing the absolute shear and vibration time series to their 0.5 Hz low-passed records. When the ratio between

    the absolute and the low-passed time series exceeded 9 (8) for the shear (vibration), $\pm 0.04$ seconds centered at the spike were

    replaced by the value averaged over $\pm 0.5$ seconds before and after the spike.

*- (L.135) 'calculating the shear frequency spectrum': this is probably averaging of the 8 spectra and correcting for the windowing loss; or is there more?*

The 1-s second fft length is overlapped by 50%. For each 8-s record, we average 15 spectra and correct for the windowing loss. We now specify in the text that the 1-s fft length is also half overlapping, and that the final spectrum shown for the 8-s portion is an average.

    half overlapping 8-s long portions, corresponding to roughly 10 m portions along the transects. A fast Fourier transformation

    (FFT) length corresponding to 1 s was chosen, and each half overlapping 1-s segment was detrended, and smoothed using a

    Hanning window, before averaging them to get the shear frequency spectrum for each 8-s segment. Spectral loss due to the size

145    of the shear probe was corrected for.

*- (L.150) The underlying assumption to exclude the larger shear sensor value if surpassing a factor of 5 is probably that the previous despiking was not perfect, and the factor 5 is from experience. Is there a rationale for this, e.g. 'at 8s-segments a factor 5 anisotropy between horizontal and vertical is so rare, that we can safely assume a particle collision happened to the sensor with larger signal'?*

Thank you for pointing to this. The reason for using a factor of 5 is, as you say, based on experience, and to simply allow for a security net when one noisy estimate from a probe survived the despiking. This threshold is routinely implemented in our processing, however, for this dataset, the two probes always agreed to within a factor of 4. We now removed the mention of factor of 5 to avoid confusion, and state that the final dissipation estimate is an average of the two probes, and that they always agreed to within a factor of 4.

160   estimated $\varepsilon$ is close to the full integration. When using two shear probes, the dissipation rate in the segment is calculated as the average of the values from both sensors. In our data, the two sensors always agreed within a factor of 4.

*- (L.156) Is there a criterion why change rates of 10° roll/s, 5° pitch/s, and 2 RPM/s have the same consequence?*

The final screening done here is based on visual control of the time series, where an abrupt rate of change of flight behavior can lead to contaminated dissipation estimates. The chosen thresholds delineated these and are used to remove outliers. We've produced scatter plots of the rate of change vs dissipation and decided on the rate of change thresholds based on those. A change in roll is generally a much smaller disturbance on the shear sensor than a change in pitch (because of how the MR is mounted on the AUV), thus a higher value is allowed. There is no criterion why the different thresholds on the different variables should have the same consequence.

We now specify in the processing description that the thresholds in the final screening are based on visual inspection of the rate of change vs dissipation estimates.

165   when the flow over the shear probe is no longer laminar (Osborn and Crawford, 1980). Final data screening excluded data with rate of change exceeding 10, 5 and 2 units per 1 s for roll, pitch and RPM, respectively. The thresholds in the final screening was determined from visual inspection of rate of change versus dissipation estimates.

*For chapter 4:*

*- (L.171f) The cleaned spectra additionally show extra removal of signal. Fig.4 shows a factor 1.5 to 2 reduction in the frequency band from 1 to 8 Hz, although there is no relevant vibration. Similar for the band between 25 and 50Hz. What is the reason for this? And what is the bias in estimated epsilon caused by this?*

Thank you for pointing this out. Indeed, the cleaned spectra show an extra removal of signal even when vibration is not relevant. This is because the Goodman algorithm relies on the squared-coherency between signals produced by shear probes and vibration sensors, thus is always bigger than zero even if signals are completely incoherent. We did not correct for this bias. The bias (in the integrated shear variance) can be corrected by dividing the spectrum by $1 - \frac{1.02 N_v}{N_f}$ where $N_v$ is the number of vibration signals used and $N_f$ is the number of fft-segments used. This result can be obtained from Nuttall, A., 1971: Spectral estimation by means of overlapped fast Fourier transform processing of windowed data. NUSC Tech. Rep.

5291, [Available online at https://apps.dtic.mil/dtic/tr/fulltext/u2/a182402.pdf], Naval Underwater Systems Center; Lueck (2022, personal communication). In our case $N_v$ and $N_f$ are 2 and 15 respectively, which would increase the spectra by a factor of about 1.16. The figure below shows an example where we correct the cleaned spectra by a factor of 1.16. This is less than the factor 1.5 to 2 you mention, and we assume the remaining difference between the uncleaned and cleaned signal could be accounted for by the removal of noise even if the vibration signal is relatively low.

[Figure]

The resulting bias on the dissipation estimate can be slightly larger, carried over from the estimate of upper cutoff wavenumber and other processing steps. A comparison of the Nasmyth spectrum multiplied by a factor of 1.16 and the Nasmyth spectrum calculated from higher epsilon, shows that a factor of 1.16 offset in the spectrum translates to an increase in epsilon by a factor of 1.22 to produce the same spectrum. Hence the Goodman bias has a slightly higher impact on the epsilon estimates than on the spectra, but this is not systematically quantified here and can be nonlinear. We did not reprocess our data set to remove the bias induced by the Goodman algorithm; however, we will adopt this correction when the method is scientifically reported. We are aware of a study in progress led by Rolf Lueck on this topic.

We now comment on the bias introduced by using the Goodman method in the results and elaborate on this in the discussion (see a snapshot of the discussion below):

245      The method for noise removal relies on the squared-coherency between the shear probe signal and the accelerometer signals (Goodman et al., 2006). Removal of the shear probe signal coherent with the accelerometer signal produces clean spectra (Figure 5). The clean spectra show that the spikes in the 10-30 Hz band have been successfully removed. Note, however, that squared-coherency will always be nonzero even when the shear probe and accelerometer time series are completely incoherent. This results in a bias by removing some of the incoherent signal. We did not correct this bias in our study. When we apply a

250    simple correction recommended in the ATOMIX guidelines, the clean spectra increase by a factor of about 1.2. This potential bias does not affect our conclusions.

*For chapter 5:*

*- (L.214) Is it clear that vehicle vibrations are the source of the 10-30Hz frequencies? If yes, integrating the MR into the AUV instead of using brackets (as proposed in L.239ff and L.250f) would not have much effect. What is the contribution of the brackets, of the MR body, of the shear sensor shafts? Only the part of the brackets possibly could be remedied.*

The phrase "Vehicle vibration" was inaccurate in our paper. We also mean to include vehicle motion such as roll and pitch. One of the major issues with mounting the MR below the AUV as we did, is the reduction in AUV maneuverability. The additional off-center drag due to the MR caused more pitch than during a dive without the MR, and subsequently, the AUV rudders made more adjustments than during a dive with no MR. This behavior could be reduced (if not avoided) by mounting the AUV in a more streamlined fashion. We cannot quantify the individual contributions to vibration from the brackets, the MR body, or the shear sensor shafts.

The phrase "vehicle vibration" is now clarified in our discussion, and we elaborate on the contamination in the 10-30Hz band (inserted below).

> from the three-bladed propeller operating at 1500 RPM. While vibrations from the propulsion system are less than ideal for turbulence measurements, the contamination occurs in a narrow band and is easily detected by both the accelerometers. The
>
> 240 vibrations detected between 10 and 22 Hz (9–20 cpm) are more worrisome as this contamination covers a broader band of the turbulence spectrum, in the wavenumbers where the spectrum typically rolls off. The spectral peaks in the shear spectra are at different frequencies for $\partial w/\partial x$ and $\partial v/\partial x$, suggesting that the vehicle motion (pitch, roll, yaw) is the main source of this contamination. While $\partial v/\partial x$ will be affected by the roll and yaw fluctuations, $\partial w/\partial x$ will be affected by changes in pitch but should be fairly insensitive to changes in roll and yaw.

*- Fig.5: Suddenly the theoretical Panchev-Kesich spectrum pops up, only in the left panel, without having been introduced before. Its only usage in the paper is in L.218 stating that the spectra resemble both Nasmyth and Panchev-Kesich. However, inspecting Fig.5a, it seems the cleaned spectra fit the Panchev-Kesich spectral shape much better than Nasmyth. (Only the estimated epsilon would result a little lower after a thorough fit.) If so, much of the discussion on early rolloff and spectrum averaging (L.219-224) would be obsolete (L.184 in Results is affected, too). Instead, the interesting question could be discussed why Panchev-Kesich shows the more similar spectral shape.*

Thank you for pointing this out. The Panchev-Kesich spectrum is now introduced earlier in the Results section and is now included in both panels of Figure 5 (see attached figure below). In addition, we now produce Panchev-Kesich spectra following the same averaging as for the Nasmyth spectra. In the original version, the Panchev-Kesich spectrum was presented as a reference, plotted using the average dissipation value (hence the roll-off was not smeared out as it would be after averaging over several spectra covering one decade range of dissipation values). Now, the individual Panchev-Kesich spectra (from the individual epsilon estimates) are bin-averaged similar to the Nasmyth spectra and shear spectra. As you point out, the roll-off in our observations does fit the Panchev-Kesich spectra better than the Nasmyth spectra. Yet there is still a systematic offset at the roll-off, and we think this is worth some consideration, hence retain a modified version of the discussion in line 219-224. For wavenumbers between 1 and 10, the Nasmyth spectra fit our observations better for epsilon>1e-7. A discussion about similarities between our observations and the two spectra is now included (inserted below).

[Figure]

The bin-averaged shear spectra suggest that the most energetic wavenumbers are not fully resolved by our instrumentation; i.e., the transition between the inertial subrange and the dissipation subrange rolls off at lower wavenumbers compared to the similarly bin-averaged Nasmyth and Panchev-Kesich spectra (Figure 6). While the shape of the spectral roll-off is relatively

255    similar to that in the Panchev-Kesich spectrum, the offset between the observed and the theoretical spectra is significant. The bias unaccounted for in application of the Goodman method cannot explain this offset fully. Some of the discrepancies in

the roll-off are likely caused by averaging the spectra over variable dissipation rates, where the spectral peak shifts to higher wavenumbers with increasing $\varepsilon$, which will smooth the spectral roll-off. Yet, to mimic the effect of smoothing of the spectral roll-off, we bin-average individual Nasmyth and Panchev-Kesich spectra similarly. Another possible reason for the difference

260    between the observed and the Nasmyth spectra is that an unknown fraction of the removed shear probe signal coherent with the accelerometers can be natural turbulence, indistinguishable from the vibrations caused by the AUV (Palmer et al., 2015). This likely leads to a reduction of variance in the contaminated band between 10–30 Hz (9–27 cpm).

*(3)*

*- The expected uncertainty of resulting epsilon is not stated/estimated. The resulting effect of basic MR uncertainty plus vehicle noise plus post processing in sum will probably exceed the typical factor 2 for calm platforms.*

Thanks for bringing this up. A recent study (Lueck, R.G., 2022: The statistics of turbulence measurements. Part 2: Shear spectra and a new spectral model. J. Atmos. Oceanic Tech., submitted) estimates the 95% confidence interval around a mean spectrum as the factor

$e^{\pm 1.96 \times \frac{5}{4}\left(N_f - N_v\right)^{-\frac{7}{9}}}$ where $N_f$ is the number of fft segments and $N_v$ is the number of vibration signals. Hence, the 95% confidence interval around a mean spectrum in our data with $N_f$ = 15 and $N_v$=2 is about [0.72 1.40]. The 95% confidence interval carried over to our epsilon

estimates, including an additional 10% sensor sensitivity calibration uncertainty, becomes about [0.6 1.6]. We now include this information in the results (inserted below).

180    shown. The 95% confidence interval around a mean spectrum can be calculated as the factor $\exp(\pm 1.96 \times \frac{5}{4}(N_f - N_v)^{-\frac{7}{9}})$,

where $N_f$ is the number of fft-segments and $N_v$ is the number of vibration signals (Lueck, 2022). Using $N_f = 15$ and $N_v = 2$, we obtain [0.72 1.40]. The 95% interval carried over to our epsilon estimates, including an additional 10% sensor sensitivity calibration uncertainty, becomes about [0.6 1.6].

*- The comparison between the three epsilon timeseries and the two single vertical MSS profiles (Fig. 7c) seems questionable. AUV epsilon is extremely variable from noise level to very high values of 5e-6. The two MSS profiles differ a factor of about 50, one is near AUV noise level, one is near 1e-6. Trying to compare statistically would mean that we'd have to check the hypothesis if the two MSS values can stem from the AUV epsilon distributions. This hypothesis will certainly not be rejected, but: the question is, which imaginable MSS profiles would be rejected at all? As the basic distribution spans more or less the entire possible range, nearly any MSS measurement would not contradict the zero hypothesis. Two noise level profiles would not; two profiles of 5e-6 would not; maybe two profiles of 1e-5 would have cast doubts. That means that the two MSS profiles probably cannot support the statement 'dissipation from AUV agrees well with MSS profiles' (L.8ff, L.206f, L.247f) in a valuable manner, even if they don't contradict either. I'd recommend to comment the comparison (L.8ff, L.206f, L.247f) more cautious. The MSS profiles might confirm the high variability of epsilon in the region, and they might constrain the uncertainty of the AUV measurements to a factor of 5 or 10.*

Thank you for addressing this. We agree with you on all accounts. The only way the two MSS profiles could potentially contradict the zero hypothesis is if they were off by a factor of 50 or more on the low epsilon side (say both profiles estimated 1e-8), suggesting the region was not as turbulent as the MR measured. We have rewritten this comparison, and now make sure to point out the shortcomings of such a comparison in the results (inserted below).

smaller at greater depth, which is expected in the boundary layer. The arithmetic mean (including 95% confidence intervals)
225    of the natural logarithm of the dissipation rates along the horizontal transects are compared to vertical microstructure profiles (Figure 8c). Although the spatial variability of $\varepsilon$ is known to be large (Yamazaki et al., 1990), the vertical profiles and the horizontal transects show comparable dissipation rates. Note however, that the comparison between the two MSS profiles and the average dissipation rates must be interpreted with cation. The two MSS profiles differ by one order of magnitude, enveloping the AUV-based measurements, and cannot be used to statistically test the validity of the AUV measurements.

*Minor comments:*

*L.27f: '... the traditional methods limit the spatial and temporal coverage of the measurements.' Better: '... limit the horizontal and temporal resolution ...', if this was meant.*

Revised as: "However, the vertical profiling limits the horizontal and temporal resolution of the measurements.".

*L.15-41: For better readability I'd propose to put L.23&24 to the previous paragraph, and to lump L25-31 into one paragraph. Thus there'd be 3 paragraphs: on turbulence measurement in general; on traditional and robotic platforms; on vibrations of robotic*

*platforms.*

We agree that it improves the readability and made the changes as you suggested.

*Fig.1a needs more clarity. Isobaths may be negligible. The ice edge is unnecessarily hard to spot, a thicker black line with an overlying dashed white line might be a solution. The experiment location should not be directly on the edge of the plot; expanding to 40°E at least would better support orientation.*

Agreed. Figure 1 is updated as shown below.

[Figure]

*L.82-91: The entire paragraph about the particulars of reckoning the front location would fit a scientific paper on observing the frontal zone. For the purpose of this technical note, it could be distilled to a single sentence, saying that the AUV was programmed to follow the frontal zone and did this successfully.*

Agreed. The entire section has been distilled and attached to the previous section. See snapshot of section 2.1 shown earlier.

*L.92: replace 'MicroRider' by 'Turbulence package' (the term MicroRider is first introduced in L.93)*

Agreed. We changed the title of section 2.2 to "Turbulence package".

*L.140: 'the shear probe signal ... was removed' sounds as if the raw shear timeseries was filtered before calculating the spectra*

Indeed. We now write "the shear spectrum signal coherent with the accelerometer spectrum signal was removed…"

*Equation (4) is a bit misleading, the second '=' is not correct. 'epsilon = left hand side of eq.4' is the exact equation, while 'epsilon estimated = right hand side of eq.4 plus correction for unresolved wavenumbers' is the practically used equation, trying to be as close to the exact equation as possible.*

Thank you for pointing this out. We now changed our equation as shown below.
In addition, we point out in the text that epsilon in the unresolved part of the spectra is corrected for by using the Nasmyth spectra, so that our final estimate is close to the full integration.

$$\epsilon = \frac{15}{2}\nu\overline{\left(\frac{\partial v}{\partial x}\right)^2} = \frac{15}{2}\nu\int_0^\infty \Psi(k)dk \approx \frac{15}{2}\nu\int_{k_1}^{k_u} \Psi(k)dk \qquad (3)$$

155 where $\nu$ is the kinematic viscosity and overbar denotes averaging in time (e.g., Fer et al., 2014). The lower ($k_1$) integration limit is determined by the wavenumber corresponding to the FFT length, and the upper ($k_u < \infty$) integration limit is usually determined from a minimum in a low-order polynomial fit to the wavenumber spectrum in log-log space. Typically electronic noise takes over after the minimum in the spectrum. To account for the variance in the unresolved part of the spectrum (integration outside the $k_1$ and $k_u$ limits), the empirical model for turbulence spectrum determined by Nasmyth (1970) was used, hence the

160 estimated $\varepsilon$ is close to the full integration. When using two shear probes, the dissipation rate in the segment is calculated as the

*L.164: Is there a conceivable reason why the AUV should be 10% faster in 0°C water than in 1°C water? If true, what would that mean for missions in the tropics?*
No, this is only because we set the AUV to use a constant RPM. The front is highly dynamic, and it is not surprising if the current changes direction across the front.

*L.166: The main energy should be at 75Hz (the propeller having 3 blades), and indeed Fig.4 points to this. However, the system seems not to be perfectly symmetric, showing 25Hz and 50Hz as well.*
Thank you for pointing to this. The propulsion system is not perfectly balanced around its rotational axis, and vibrations at 25, 50 and 75 Hz (and the harmonics of these frequencies) are induced by this off-center rotation. The main energy is indeed at 75 Hz, related to the 3-bladed propeller. We now include this information in our results (inserted below).

An RPM of 1500 corresponds to 25 Hz, or using a mean speed of $1.1 \, \mathrm{m\,s^{-1}}$ to 23 cycles per meter (cpm). The contamination

185 of the shear spectra by the propulsion system is visible in Figure 5; the propulsion system is not perfectly balanced around its rotational axis, and vibrations at 25, 50 and 75 Hz (and the harmonics of these frequencies) are induced by this off-center rotation. The main contaminating energy is at 75 Hz, related to the 3-bladed propeller. In addition, the accelerometers indicate

*L.169: 'by the propeller at 25Hz'. Better 'by the propulsion system at 25Hz' (see previous remark). Delete 'of the propeller frequency' for the same reason.*
Agreed. We corrected as you suggested (see snapshot above).

*L.180 to 186: all remarks in parentheses should be deleted. This is information for the figure caption*
Agreed.

*L.190: what is the purpose of this sentence? The important differences between sensor 1 and 2 discussed in the following are systematic, while the fact that the single epsilon values are calculated from individual spectra only explains scatter.*
We agree that this sentence is unnecessary. We rewrote the start of this paragraph.

*L.197: 'The distribution is not log-normal'. I'd recommend to delete the sentence. Probably none of the distributions shown is log-normal; and the next sentence 'A second mode appears ...' already implies that Fig.6e is not log-normal.*
Agreed. We deleted this sentence.

*L.204: 'boundary layer': please state at some point in the paper (maybe the beginning of chapter 2) how deep the mixed layer is.*

The mixed layer was about 60 m at the time and location of the mission. We now include a short description of the environmental conditions in the beginning of chapter 2 (lines 67-68, inserted below).

after completing three crossings of the front. Before and during the AUV mission, the wind speed was around $10\,\mathrm{m\,s^{-1}}$, air temperatures were close to -5°C measured at $15\,\mathrm{m}$ height, and the surface boundary layer extended to about $60\,\mathrm{m}$ depth.

*L.204: 'maximum likelihood estimate': the arithmetic average should do. Or is this what you mean?*

We used the maximum likelihood estimate but have now changed this to the arithmetic average as suggested (inserted below). Note, we used the arithmetic mean of the natural logarithm of the dissipation rates, which is the same as the geometric mean of the dissipation rates.

[Figure]

**Figure 8.** Overview of dissipation rates. Time series of **a)** temperature and **b)** final estimate of the dissipation rate. **c)** Vertical profiles (black and green) of $\varepsilon$ measured by the vertical microstructure profiler at two co-located stations marked by stars in Fig. 1b. Mean value and the lower and upper limits (95% confidence intervals) of the natural logarithm of the AUV-MR measurement are shown at their corresponding average depth in (c). Blue, red and yellow corresponds to 11, 21 and 31 m depth respectively.